# Clinical Determinants Predicting *Clostridioides difficile* Infection among Patients with Chronic Kidney Disease

**DOI:** 10.3390/antibiotics11060785

**Published:** 2022-06-08

**Authors:** Łukasz Lis, Andrzej Konieczny, Michał Sroka, Anna Ciszewska, Kornelia Krakowska, Tomasz Gołębiowski, Zbigniew Hruby

**Affiliations:** 1Research and Development Center, Department of Nephrology, Provincial Specialist Hospital, Kamienskiego 73a, 51-124 Wroclaw, Poland; lislukasz@ymail.com (Ł.L.); michal.sroka@wssk.wroc.pl (M.S.); zhruby@wssk.wroc.pl (Z.H.); 2Department of Internal Medicine, University Hospital, Witosa 23, 45-401 Opole, Poland; 3Department of Nephrology and Transplantation Medicine, Wroclaw Medical University, Borowska 213, 50-556 Wroclaw, Poland; korneliakrakowska006@gmail.com (K.K.); tomasz.golebiowski@umw.edu.pl (T.G.); 4Department of Intensive Care and Anesthesiology, Provincial Specialist Hospital, Kamienskiego 73a, 51-124 Wroclaw, Poland; aniaa.ciszewska@gmail.com; 5Department of Clinical Nursing, Wroclaw Medical University, Bartla 5, 51-618 Wroclaw, Poland

**Keywords:** *Clostridioides difficile*, chronic kidney disease, malnutrition, pseudomembranous enterocolitis

## Abstract

The majority of recently published studies indicate a greater incidence rate and mortality due to Clostridioides difficile infection (CDI) in patients with chronic kidney disease (CKD). The aim of this study was to assess the clinical determinants predicting CDI among hospitalized patients with CKD and refine methods of prevention. We evaluated the medical records of 279 patients treated at a nephrological department with symptoms suggesting CDI, of whom 93 tested positive for CDI. The survey showed that age, poor kidney function, high Padua prediction score (PPS) and patients’ classification of care at admission, treatment with antibiotics, and time of its duration were significantly higher or more frequent among patients who suffered CDI. Whereas BMI, Norton scale (ANSS) and serum albumin concentration were significantly lowered among CDI patients. In a multivariate analysis we proved the stage of CKD and length of antibiotics use increased the risk of CDI, whereas higher serum albumin concentration and ANSS have a protective impact.

## 1. Introduction

*Clostridioides difficile* infection (CDI) is caused by a Gram-positive, anaerobic, spore-forming bacillus, the most prevalent cause of a nosocomial diarrhea worldwide [1]. It is transferred by a fecal–oral route and can have either a mild course or progress to a life-threatening colitis, with diarrhea, abdominal pain, dehydration, fever, and subsequent circulatory shock [2].

Antibiotic treatment, older age, and hospitalization belong to the most significant risk factors for CDI [3,4]. The other well-defined clinical conditions, predisposing to CDI, include an inflammatory bowel disease, malignant tumors, transplantations, and chronic kidney disease (CKD) [3,5].

The influence of proton pomp inhibitors (PPI) on the incidence of CDI remains controversial. Several studies have found a significant association between PPI treatment and CDI [6,7]; however, there are also a number of papers where such correlation was not proven [8,9].

Although the estimated burden of *Clostridioides difficile* (CD) health care-associated infections decreased in the United States by an adjusted 24% from 2011 to 2017 [10], it has been still recognized as a leading cause of infection among hospitalized patients and a considerable threat to public health globally [11].

Some of the most vulnerable patients are those suffering from CKD and in particular with end-stage renal disease (ESRD), despite the implementation of CDI prevention strategies [12]. The majority of recently published studies indicate a greater incidence rate and mortality due to CDI in CKD, especially among those with ESRD, in comparison to the general population [13,14,15]. It also results in a significant increase in the treatment costs and prolonged hospitalization time [16].

The main aim of this paper was to assess clinical determinants for predicting CDI among hospitalized patients with CKD and refine methods of prevention to combat the epidemic of nosocomial infections with CD etiology.

## 2. Materials and Methods

This was a single center, retrospective study, including data of 15,389 patients hospitalized in a department of nephrology, between 1 January 2016 and 31 December 2020, who developed symptoms indicating CDI. A flowchart presenting initial qualification, the screening of patients, and assignment to CDI positive and CDI negative groups is presented in Figure 1.

Qualifying symptoms were diarrhea (>3 stools per day) and abdominal pain and/or fever [2]. Although we based on definition of CDI provided by the Infectious Diseases Society of America (IDSA) and Society for Healthcare Epidemiology of America (SHEA), we included only patients who developed the symptoms within at least 72 h after admission.

In all patients a rapid enzyme cassette immunoassay was performed, detecting the antigens of toxins A and B of CD in stool (TOX A/B QUIK CHEK^®^; Techlab, Blackburg, VA, USA).

The exclusion criteria were missing clinical data, length of stay (LOS) shorter than 3 days, or admission from another hospital.

The following data were assessed: patients’ age; gender; body mass index (BMI); presence of concomitant diseases; length of stay (LOS); stay in an emergency department (ER) directly before admission; presence of acute kidney injury (AKI), defined according to the KDIGO [17]; and pharmacotherapy with the emphasis on antibiotics, proton pump inhibitors (PPI), statins, probiotics, and immunosuppression.

At admission each patient was assessed using the Norton scale (ANSS) and the classification of patient care, evaluated by the ward nurse, and the Padua prediction score (PPS), assessed by the physician.

ANSS assesses the risk of pressure sores during hospitalization. It consists of five variables: physical and mental condition, activity, mobility, and incontinence. Each domain is graded from 1 to 4 points and final admission ANSS ranges between 5 and 20 points and an ANSS ≤ 14 is considered as being low [16].

The classification of patient care is a clinical tool used for managing and planning the allocation of nursing staff in accordance with the nursing care needs. It is subdivided into four classifications, namely: 1—self-care patient, 2—partial care patient, 3—complete care patient, 4—critical care patient.

The PPS identifies admitted patients at a high risk for venous thromboembolism (VTE) and who would benefit from thromboprophylaxis. In the PPS, the risk profile for VTE is calculated using 11 common risk factors [18]. Each risk factor is weighted according to a point scale. A high risk of VTE is defined as a cumulative score ≥ 4 and a low risk as < 4 [18].

Laboratory tests were performed in the hospital laboratory using standard methods, including the concentration of serum creatinine (sCr), serum urea, and albumin (ALB). The shortened Modification of Diet in Renal Disease (MDRD) equation was used to calculate the estimated glomerular filtration rate (eGFR) [18].

Statistical analysis was performed utilizing the STATISTICA ver. 12 software (StatSoft Inc., Tulsa, OK, USA). Numerical data were expressed as mean and standard deviations (SD). Normal distribution was verified with the Kolmogorov–Smirnov test, which enabled the assessment of the differences between the two groups with Student’s t test, the homogeneity of variations being checked with Fisher’s test. In case of non-linear distribution, statistical importance of the differences was evaluated with the use of the U Mann–Whitney test. For quantitative data, χ2 analysis was performed. The influence of the parameters of CDI occurrence was tested with the implementation of logistic regression. The statistical significance cut-off level was set at *p* = 0.05.

## 3. Results

### 3.1. Patients’ Baseline Characteristic

A total number of 279 patients, aged 68 years, 124 (44%) women and 155 (66%) men, were enrolled in the study, of whom 93 (33%) had proven CDI and 186 were without CDI. All patients presented symptoms suggesting CDI, e.g., diarrhea and abdominal pain, whereas 45 in group of CDI positive patients and 76 of CDI negative patients had fever additionally.

### 3.2. Differences between CDI and Control Group

Patients who suffered from CDI were significantly older and displayed poor kidney function. They were more frequently treated with PPIs and antibiotics, for a significantly prolonged time. Moreover, they presented with higher PPS and patients’ care classifications at admission and were more frequently hospitalized in the ER before admission. Patients with CDI had higher mortality and required longer LOS. CDI-patients presented lower albumin concentration and ANSS.

It was not revealed whether treatment with statins, immunosuppression, probiotics, or the presence of diabetes, neoplasm significantly affects the risk of CDI. Comparison of differences in clinical data are presented in Table 1.

### 3.3. Univariate Logistic Regression Predicting CDI

Using univariate logistic regression models (as shown inTable 2), we have found that age, CKD stage, both serum creatinine and urea concentrations, number of antibiotics used in therapy, time of treatment, assessment in PPS, and higher patients’ care class significantly increased the risk of CDI. Whereas serum albumin concentration at admission, ANSS, and BMI lowered the risk of CDI.

In a multivariate model, CKD stage and the length of antibiotics treatment had a significant impact on CDI, whereas albumin concentration and Norton score lowered the risk, as presented in Table 3.

## 4. Discussion

Our study showed that age, declined kidney function (expressed by both serum creatinine and urea concentration and subsequently by CKD stage), higher PPS, patients’ care classification at admission, treatment with antibiotics and length of its duration were significantly higher or more frequent among patients who suffered from CDI. Whereas BMI, ANSS, and serum albumin concentration were significantly lower among CDI patients.

In a multivariate analysis both the stage of CKD and length of antibiotics use increased the risk of CDI, whereas higher serum albumin concentration and ANSS lowered the CDI risk. These factors are the best clinical determinants for predicting the presence of CDI among patients with CKD. No effect was found for other factors, including treatment with statins, immunosuppression, probiotics, or presence of diabetes and neoplasm.

Age is the most frequently reported risk factor for CDI [12,13]. It is associated with an increased number of comorbidities, malnutrition, and reduced psychomotor skills. These may result in a lower ANSS (≤14 is considered as being low) and higher PPS (high risk of VTE is defined as a cumulative score ≥ 4) or care classification (III—complete care patient and IV critical care patient), which were proven to be significant factors for CDI, to our knowledge probably first time in the literature.

These findings might be applied to screen the most vulnerable patients and to shorten their hospitalization, ER stay, or time of laboratory tests and to increase vigilance of medical personnel for aseptic behavior. The ANSS, PPS, and care classification is easy to learn, easy to use, and, most importantly, it is already being used successfully throughout the world to assess the risk of pressure ulcers, venous thromboembolism, or patient nursing care needs.

It is worth underlining that ER stays before admission were an independent risk factor for CDI patients with CKD in our survey. Moreover, in one of recent study it has been suggested that ER may be one of the main reservoirs of CDI [19].

Lower BMI and albumin concentration were definitive clinical determinants for predicting CDI. This fact has been confirmed in other papers and is associated with malnutrition and secondary immunodeficiency due to deproteinization [15,20]. Bearing in mind that lower albumin concentration was one of the best determinants for predicting CDI in our study, improvements in diagnosing and treating hospital malnutrition are needed, the effects of which could benefit both patients and healthcare providers.

It has been reported that the use of antibiotics and, in particular, the length of antibiotic treatment significantly increased the risk of CDI. Antibiotics policy seems to be the most important factors, influencing a significant reduction in CDI frequency. The importance of the problem is crucial because it is estimated that approximately 50% of antibiotics used in hospitals are considered unnecessary [21].

Most studies have confirmed that PPIs increase the risk of CDI. The risk is estimated to be 1.7–2.3 times higher [6,7]. Given the widespread abuse of the above-mentioned drugs, caution in their use seems to be indispensable.

Patients with advanced stage of CKD (especially those in the ESRD phase) and with the presence of AKI at admission were also at risk of CDI, according to our results. Several studies have found a significant association between advanced CKD, AKI, and CDI [22,23,24], but, on the other hand, there are papers where such correlation was not proven [25,26]. In our previous study, it was not documented that reduced eGFR augmented the risk of the CDI [15]. This may be attributable to the fact that the control group in this study consisted of CD-negative patients with diarrhea and the investigated group of patients was dominated by those with class 5 CKD: 137 of 207 (65.7%) with 77 (37.2%) of them chronically dialyzed. On the other hand, in our study, the control group consisted of 186 patients with signs of infection but without CDI, who were admitted to the same department and hospitalized at the same time as patients with CDI. The investigated group was not dominated by patients with CKD class 5: 128 of 279 (45.8%) with 82 (29.4%) of them undergoing dialysis.

It was also found that patients with CDI had higher mortality and required longer LOS and most of recent publications have confirmed that correlation. Pant et al. showed that if the average duration of hospitalization is longer than 9 days, then its costs rise additionally on 68 thousand dollars, and mortality is twice as high [16,22].

Clinicians should be aware of these clinical determinants predicting CDI in CKD patients, because some of them are modifiable and amenable to effective interventions. Special attention should be devoted to the rapid diagnosis of CDI and rational antibiotics policy, aimed at reducing the use of unwarranted antibiotic therapy, avoiding drugs increasing the risk of CDI, and shorten the time of treatment duration. Furthermore, aseptic behavior, the proper nutrition of malnourished patients; systematic education and control of medical personnel; and cautious use PPIs, limiting them to situations where they are necessary, especially in patients with low albumin concentration, ANSS, and advanced CKD, could significantly reduce CDI-associated morbidity and mortality among adults, particularly those with CKD.

Our study has some limitations. Firstly, all patients who were enrolled in the study were of Caucasian origin. Secondly, analysis was based on patients’ data over a 5-year period. The survey relies only on the single center experience. To provide a robust clinical tool, allowing the identification of individuals at high risk of CDI among CKD patients, a long-term multicenter study, including larger cohort, is required.

## 5. Conclusions

The best clinical determinants predicting the presence or absence of CDI among patients with CKD are stage of CKD and the length of antibiotics use, increasing the risk of CDI, whereas higher serum albumin concentration and ANSS have a principal protective impact.

## Figures and Tables

**Figure 1 antibiotics-11-00785-f001:**
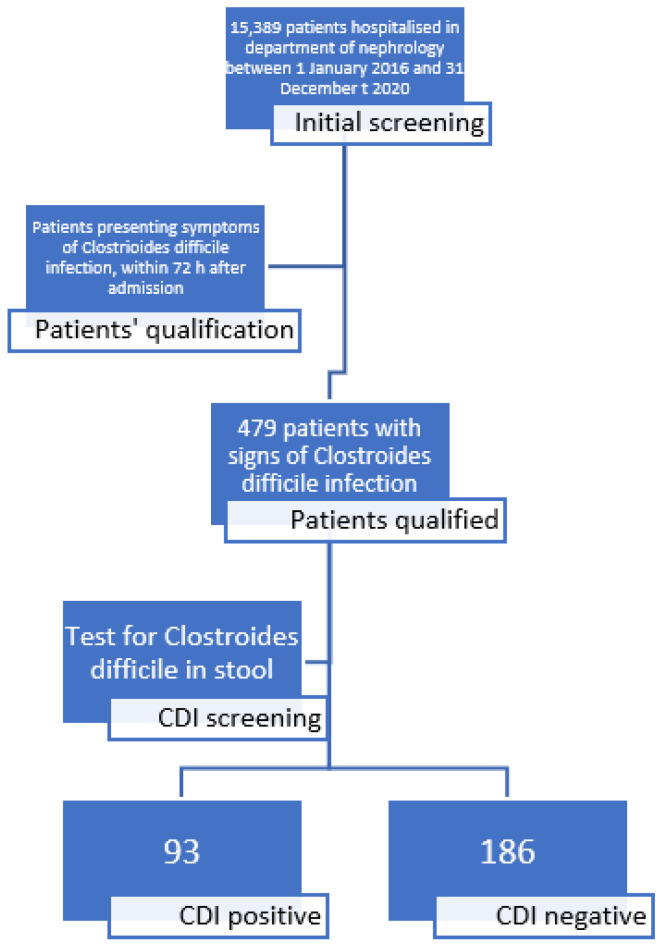
Patients’ screening and recruitment.

**Table 1 antibiotics-11-00785-t001:** Comparison of clinical parameters between CDI-patients and non-CDI.

Parameter	CDI (N = 93)	Non-CDI (N = 186)	*p* * Chi^2^-Test
Age [years]	72.1 ± 13.8	65.6 ± 16.1	0.001
BMI	23.6 ± 5.6	26.7 ± 4.6	<0.0001
LOS [days]	30.7 ± 18.5	8.9 ± 6.3	<0.0001
sCr at admission [mg/dL]	3.8 ± 3.9	2.3 ± 1.7	0.0002
Urea concentration at admission [mg/dL]	144.6 ± 102.6	84.2 ± 56.1	<0.0001
CKD stage	4.3 ± 1.1	3.6 ± 1.4	<0.0001
HD treatment	36 (39%)	46 (25%)	0.016 *
ALB at admission [g/dL]	2.8 ± 0.6	3.7 ± 0.5	<0.0001
Use of antibiotics	89 (96%)	54 (29%)	<0.0001 *
Number of antibiotics used	2 ± 1	0.4 ± 0.7	<0.0001
Length of antibiotics treatment [days]	15.7 ± 8.7	2.6 ± 4.4	<0.0001
PPS	4.6 ± 1.9	1.6 ± 1.5	<0.0001
ANSS	12.5 ± 3.3	17.5 ± 2.2	<0.0001
Patients’ care class 1/2/3/4	2.7 ± 0.5	1.8 ± 0.6	<0.0001
Presence of neoplasm	11 (12%)	18 (10%)	0.6 *
DM	29 (31%)	52 (28%)	0.58 *
PPI treatment	65 (70%)	92 (49%)	0.002 *
Use of probiotics	42 (45%)	14 (8%)	<0.0001 *
Use of statins	24 (25%)	63 (34%)	0.17 *
Immunosuppression use	17 (18%)	47 (25%)	0.19 *
Death	18 (19%)	9 (5%)	0.0001 *
ER stay	89 (96%)	71 (38%)	<0.0001 *
AKI at admission	35 (38%)	19 (10%)	<0.0001 *

Abbreviations: LOS—length of stay; CKD—chronic kidney disease; sCr—serum creatinine concentration; ALB—serum albumin concentration; CDI—Clostridioides difficile infection; HD—hemodialysis; PPI—proton pump inhibitor; DM—diabetes mellitus; ER—emergency department; AKI—acute kidney injury; ANSS—the Norton scale; PPS—the Padua prediction score; *—for non parametric variables Chi^2^ was applied.

**Table 2 antibiotics-11-00785-t002:** Univariate logistic regression in predicting CDI.

Variable	Estimate	Odds Ratio	95% Confidence Interval		*p*-Value
Age	0.03	1.03	1.01	1.05	0.001
CKD Stage	0.45	1.57	1.26	1.96	0.001
sCr at admission [mg/dL]	0.24	1.27	1.09	1.47	0.002
Urea at admission [mg/dL]	0.01	1.01	1.006	1.02	0.001
ALB at admission [g/dL]	−2.27	0.1	0.06	0.18	0.001
Number of antibiotics	1.91	6.8	4.4	10.4	0.001
Length of antibiotics use [days]	0.33	1.38	1.28	1.49	0.001
PPS	0.82	2.26	1.89	2.71	0.001
ANSS	−0.57	0.56	0.5	0.64	0.001
Patients’ care class	2.4	11.04	6.25	19.5	0.001
BMI	−0.14	0.87	0.82	0.94	0.001

Abbreviations: CKD—chronic kidney disease; sCr—serum creatinine concentration; ALB—serum albumin concentration; ANSS—the Norton scale; PPS—the Padua prediction score; BMI—body mass index.

**Table 3 antibiotics-11-00785-t003:** Multivariate logistic regression in predicting CDI.

Variable	Estimate	Odds Ratio	95% Confidence Interval		*p*-Value
CKD Stage	0.53	1.7	1.01	2.7	0.02
ALB at admission [g/dL]	−1.4	0.25	0.1	0.58	0.001
Length of antibiotics use [days]	0.26	1.3	1.19	1.42	0.001
ANSS	−0.39	0.68	0.57	0.82	0.001

Abbreviations: CKD—chronic kidney disease; ALB—serum albumin concentration; ANSS—the Norton scale.

## Data Availability

Data are contained within the article.

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
