# Peer review of "Clinical Determinants Predicting Clostridioides difficile Infection among Patients with Chronic Kidney Disease"

_antibiotics, 2022, doi:10.3390/antibiotics11060785_

Round 1

Reviewer 1 Report

To evaluate the predisposing factor for Clostridioides difficile infection among CKD patients, control group should be matched with case group by underlying disease such as Charlson’s comorbidity score or propensity score. However, in this study, baseline patient’s characteristics in table 2 showed significant differences. Case group showed significant high CKD stage, AKI at admission, and so on. In particular, emergency room stays mean acute and/or severe illness, more likely to use antibiotics, and more likely to be associated with other CDI risk factors. To make your thesis relevant, control groups should be matched with case group according to major risk factors – such as age, ER stay, Charlson’s comorbidity score, so on. Then you can analysis the usefulness of ANSS properly. 

As an abbreviation for proton pump inhibitors, PPI is recommended instead of IPP.

Exclusion criteria for each group mentioned repeatedly.

Tables]

Patients’ care class scored with 1/2/3/4. However, in tables, only showed 1/2/3.

Table should be modified. Table 1 and 2 should be merged or delete table 1. Table 3 and 4 also need to be merged. To show the Included variables for multivariate analysis will make a clear the analysis. 

What is meaning of ‘length of antibiotics treatment before CDI’ in control group? Control group did not develop CDI. Please define how to estimate that days.

What is meaning of ‘estimate’ in table 3&4? Please, explain in the method section.

All the parameters were mixed up. Please categorize.

All abbreviation should be mentioned full name for the first-time use. This also applied to tables.

Author Response

  1. To evaluate the predisposing factor for Clostridioides difficile infection among CKD patients, control group should be matched with case group by underlying disease such as Charlson’s comorbidity score or propensity score. However, in this study, baseline patient’s characteristics in table 2 showed significant differences. Case group showed significant high CKD stage, AKI at admission, and so on. In particular, emergency room stays mean acute and/or severe illness, more likely to use antibiotics, and more likely to be associated with other CDI risk factors. To make your thesis relevant, control groups should be matched with case group according to major risk factors – such as age, ER stay, Charlson’s comorbidity score, so on. Then you can analysis the usefulness of ANSS properly.

The inclusion criteria were defined imprecisely, and we are sorry for that mistake. All patients who during hospitalization presented signs, suggesting C.difficile infection, were included. Among them a group consisting of 93 patients had proven C.difficile infection and 186 hadn’t. Therefore, in subsequent statistics we compared those with CDI to those without it. We realized that naming the latter group as control group was misleading and we are sorry for that.

  1. As an abbreviation for proton pump inhibitors, PPI is recommended instead of IPP.

It has been corrected.

  1. Exclusion criteria for each group mentioned repeatedly.

The inclusion criteria were written imprecisely. All included patients presented diarrhea, not only those with subsequential C.difficile infection. 

  1. Patients’ care class scored with 1/2/3/4. However, in tables, only showed 1/2/3.

We added 1/2/3/4 but there was no patient in 4th class.

  1. Table should be modified. Table 1 and 2 should be merged or delete table 1. Table 3 and 4 also need to be merged. To show the Included variables for multivariate analysis will make a clear the analysis. 

Table 1 was erased. We decided to keep Table 2 (previously Table 3) and Table 3 (previously Table 4). Table 2 shows variables, which in logistic regression had predictive impact on C.difficile infection. Table 3 shows variables, which taken together have the best predictive impact on CDI prediction.

  1. What is meaning of ‘length of antibiotics treatment before CDI’ in control group? Control group did not develop CDI. Please define how to estimate that days.

This was our mistake. We meant length of treatment with antibiotics without any correlation with CDI. We sorry for that.

  1. What is meaning of ‘estimate’ in table 3&4? Please, explain in the method section.

Estimate means - logistic regression model coefficients

  1. All the parameters were mixed up. Please categorize. 

Parameters were categorized.

  1. All abbreviation should be mentioned full name for the first-time use. This also applied to tables.

It has been corrected.

Reviewer 2 Report

The authors confirm by a case-control survey the association of risk factors of chronic kidney disease with Clostridium difficile infection.

Table 1: may be eliminated, as all data are repeated in Table 2.

Table 2 and Methods: "cases" = nephrology department patients with CD infection; "controls" = nephrology department patients witout CD infection. "Cases CDI+" and "Controls CDI-" were not selected for minimal paired confounding variables such as "age", "sex", "duration of hospital stay". Due to this lack of pairing, the survey looks more like a "descriptive epidemiologic survey" than a "analytic survey". Terms "retrospective case-control study" seem abusive in this context. "Descriptive analysis of hospitalised nephrology patients with or witout Clostridium infection" would be more appropriate.

Table 3: multivariate logistic regression is forcefully significant for multiple variables as the results of Table 2 have already shown significant differences between hospitalised nephrology patients with or without CD infection. Which supplementary information does multivariabe logistic regression bring in this scenario? A scoring system based on "categories" (age < > 50 or 65 yrs; creatininin < > 1.8 mg/dL; etc.) to classify nephrology patients in "high risk" and "low risk" would have been more pertinent than a multivariate logistic regression.

Author Response

The authors confirm by a case-control survey the association of risk factors of chronic kidney disease with Clostridium difficile infection.

  1. Table 1: may be eliminated, as all data are repeated in Table 2.

Table 1 was eliminated.

  1. Table 2 and Methods: "cases" = nephrology department patients with CD infection; "controls" = nephrology department patients witout CD infection. "Cases CDI+" and "Controls CDI-" were not selected for minimal paired confounding variables such as "age", "sex", "duration of hospital stay". Due to this lack of pairing, the survey looks more like a "descriptive epidemiologic survey" than a "analytic survey". Terms "retrospective case-control study" seem abusive in this context. "Descriptive analysis of hospitalised nephrology patients with or witout Clostridium infection" would be more appropriate.

We have corrected the imprecise sentences. The control patients were those hospitalized in department of nephrology who developed signs suggesting C.difficile infection but without proof of infection. We also corrected the study description in Methods section.

  1. Table 3: multivariate logistic regression is forcefully significant for multiple variables as the results of Table 2 have already shown significant differences between hospitalised nephrology patients with or without CD infection. Which supplementary information does multivariabe logistic regression bring in this scenario?

In Table 1 (previously named Table 2) we showed differences between CDI and non-CDI groups. In Table 2 (previously named Table 3) we showed univariate logistic regression analysis, which is not exactly the same what statistical differences. This univariate analysis shows single factors which have a predictive potential. In multivariate analysis we included all parameters, which taken together have the best predictive power.

  1. A scoring system based on "categories" (age < > 50 or 65 yrs; creatininin < > 1.8 mg/dL; etc.) to classify nephrology patients in "high risk" and "low risk" would have been more pertinent than a multivariate logistic regression.

In our work we decided to build the multivariate logistic regression model, with the best predictive power, therefore we resigned from using categorized variables.

Round 2

Reviewer 1 Report

Line 58: The number, 279 should be removed. (279 is the final number of patients included in the study, not the number of initial screening subjects. However, in line 58, 279 could be mistaken for the number of subjects to be screened.) Please represent the flow diagram as a figure1.

Line 60: CD-associated enterocolitis à CDI

Line 60-61: please provide references. And please mention the data (number of patient who experience diarrhea, abdominal pain, fever) of each group at results section as clinical characteristics.   

Line 62-63: What is the definition of CDI in this study? Please define, and provide references.

If the definition is different from those of guidelines such as IDSA, it would be good to mention it as a limitation of this study.

(For example, we included patients who complained diarrhea. Among them, positive results for XXX defined as CDI patients.)

Is there any criteria or rationale to select 4 variables for multivariate analysis? I still don’t understand what this analysis means. Did you adjust age variable? CKD stage, albumin level and ANSS were evaluated at the point of admission, however, length of antibiotics occurred after admission. If CKD stage showed negative correlation with CDI development, will you explain CKD stage had protective impact on CDI development? I cannot agree with your conclusion.

ANSS is a scoring system. In conclusion, you mentioned that ‘ANSS has a protective impact.’. It is not a reasonable theory. Variables were used only to estimate the probability for occurrence of CDI. Even though high ANSS showed negative correlation with occurrence of CDI, it cannot be interpreted as a protective impact.

The errors in the table were not changed. Perhaps you misunderstood the abbreviation requirement.

-IPP in table 1.

-Full names of OR and CI should also be mentioned in table 2&3

Author Response

REVIEWER 1

  1. Line 58: The number, 279 should be removed. (279 is the final number of patients included in the study, not the number of initial screening subjects. However, in line 58, 279 could be mistaken for the number of subjects to be screened.) Please represent the flow diagram as a figure1.

I corrected this part. I have added a flowchart, as Figure 1, clarifying the process of patients’ screening and qualification.

  1. Line 60: CD-associated enterocolitisà CDI

It was corrected.

  1. Line 60-61: please provide references. And please mention the data (number of patient who experience diarrhea, abdominal pain, fever) of each group at results section as clinical characteristics.

The detailed data were mentioned in “Results” section.

  1. Line 62-63: What is the definition of CDI in this study? Please define, and provide references.

We added the reference.

  1. If the definition is different from those of guidelines such as IDSA, it would be good to mention it as a limitation of this study. (For example, we included patients who complained diarrhea. Among them, positive results for XXX defined as CDI patients.)

We based on the definition of ISDA/SHEA and tested patients with diarrhea. The limitation was that we only include those who presented those symptoms within 72 hours after admission.

  1. Is there any criteria or rationale to select 4 variables for multivariate analysis? I still don’t understand what this analysis means. Did you adjust age variable? CKD stage, albumin level and ANSS were evaluated at the point of admission, however, length of antibiotics occurred after admission. If CKD stage showed negative correlation with CDI development, will you explain CKD stage had protective impact on CDI development? I cannot agree with your conclusion.

Using logistic regression, we have built the model, with the strongest predictive power for CDI. Such model included 4 variables, e.g., CKD stage, ALB at admission, length of antibiotics use and ANSS. There were also other models including different variables but with weaker predictive power therefore we have chosen the best one. In our model CKD stage and length of antibiotics use have odds ratio >1 (1.7 and 1.3 respectively), so it means that whether CDK stage grows, the risk of infection grows also 1.7 times, and in case of every additional day of antibiotic therapy the risk of infection grows 1.3 times. Therefore, CKD class and length of antibiotics use are risk factors of CDI infection (line 154 and 155). Whereas serum albumin concentration and ANSS have OR lower than 1, so it means that every growth of both parameters lowers the risk of CDI (line 156). I have changed “protective impact” for “lowers CDI risk”.

  1. ANSS is a scoring system. In conclusion, you mentioned that ‘ANSS has a protective impact.’. It is not a reasonable theory. Variables were used only to estimate the probability for occurrence of CDI. Even though high ANSS showed negative correlation with occurrence of CDI, it cannot be interpreted as a protective impact.

The sentence was not especially precise therefore I changed it for lowers CDI risk. (line 156)

  1. The errors in the table were not changed. Perhaps you misunderstood the abbreviation requirement.

-IPP in table 1.

It has been corrected.

-Full names of OR and CI should also be mentioned in table 2&3

Full names Odds Ratio (OR) and Confidence Interval (CI) were added in the Tables 2 and 3.

Reviewer 2 Report

The remark on "case-control survey" has been taken in account and it is more appropriate to consider these observations as part of descriptive epidemiology and not of analytical epidemiology. Ok with these changes.

Redundant table I has been removed. Ok with this change.

The objective of clinical usefulness of categorizing nephrology patients in CDI high risk and CDI less elevated risk if well described and more clearly presented. Ok.

It remains to be demonstrated that a clinical strategy defining CDI risk in nephrology patients has any effect on incidence in these high-risk patients. This requires of course another long-term study with many nephrology teams. I invite you to add this limit of your observations - it does not decrease the value of your observations, but it is better to include this limit by yourself than to leave this reflection to readers. The decision of this modification depends of you.  

Author Response

  1. The remark on "case-control survey" has been taken in account and it is more appropriate to consider these observations as part of descriptive epidemiology and not of analytical epidemiology. Ok with these changes.
  2. Redundant table I has been removed. Ok with this change.
  3. The objective of clinical usefulness of categorizing nephrology patients in CDI high risk and CDI less elevated risk if well described and more clearly presented. Ok.
  4. It remains to be demonstrated that a clinical strategy defining CDI risk in nephrology patients has any effect on incidence in these high-risk patients. This requires of course another long-term study with many nephrology teams. I invite you to add this limit of your observations - it does not decrease the value of your observations, but it is better to include this limit by yourself than to leave this reflection to readers. The decision of this modification depends of you.  

 Thank you for the valuable comment. I totally agree with it and therefore I added an appropriate explanation in the Discussion paragraph.

This manuscript is a resubmission of an earlier submission. The following is a list of the peer review reports and author responses from that submission.